# How Well Can We Infer Selection Benefits and Mutation Rates from Allele Frequencies?

**DOI:** 10.3390/e25040615

**Published:** 2023-04-04

**Authors:** Jonathan Soriano, Sarah Marzen

**Affiliations:** W. M. Keck Science Department, Pitzer, Scripps, and Claremont McKenna College, Claremont, CA 91711, USA

**Keywords:** mutual information, channel capacity, Wright-Fisher model, evolution, cultural evolution

## Abstract

Experimentalists observe allele frequency distributions and try to infer mutation rates and selection coefficients. How easy is this? We calculate limits to their ability in the context of the Wright-Fisher model by first finding the maximal amount of information that can be acquired using allele frequencies about the mutation rate and selection coefficient– at least 2 bits per allele– and then by finding how the organisms would have shaped their mutation rates and selection coefficients so as to maximize the information transfer.

## 1. Introduction

Imagine the plight of an experimentalist who longs to understand the mutation rates and selection benefits experienced by a number of populations of organisms [1,2,3]. All she has is the allele frequencies over time, which she can aggregate for each population into a distribution of allele frequencies that provide information about the mutation rate and selection coefficient experienced by that population. If she were to use an unbiased estimator of the mutation rate and selection coefficient, the variance of that estimator would be lower-bounded by the inverse of the Fisher information according to the Cramer-Rao bound [4]. With some caveats [5], the Fisher information is positive correlated to the mutual information between the parameters (selection coefficient and mutation rate) and the allele frequencies.

We therefore ask, on behalf of the experimentalist: how well could she do? How could Nature maximize this mutual information between environmental factors (mutation rate and selection coefficient) and allele frequencies, and thereby allow the experimentalist to determine the mutation rates and selection coefficients as well as possible? This maximized mutual information is also known as the channel capacity [4], and the probability distribution over selection coefficients and mutation rates that achieve channel capacity provide guidance as to which environments are easiest to infer benefits and variability for. The mutual information calculated is conceptually related to, but is certainly not the same as, the mutual information between environment and phenotypes that governs the expected log growth rate of a population [6,7,8]. (Note, though, that those results [6,7,8] usually deal with setups in which the environment is fluctuating instead of setups in which separate populations are subject to different environments.) As a curiosity, then, we also ask whether or not Nature itself has made it easy for the experimentalist to estimate mutation rates and selection coefficients by comparing the optimized selection coefficients and mutation rates for the experimentalist’s inference procedure to those found in Nature.

We apply a simple and standard model of evolution, the Wright-Fisher model, to calculate this channel capacity. This forms a channel coding problem, in which mutation rates and selection coefficients might be altered so as to maximize information transfer from the environment to the experimentalist.

Our information theoretic approach is grounded in recent results that analyzed other biological species-environment channels [9,10]. These biological channels have shown to carry little information of less than two bits, including the Hunchback-Bicoid channel in the Drosophila embryo [11], tumor necrosis factor signaling [12], and a number of other pathways [13], though see work on ligand-receptor binding [14,15,16,17]. Despite the low channel capacity, the biological channel of Hunchback-Bicoid implicated in Drosophila embryo development shows evidence that the input to the channel has been optimized to help achieve maximal information transmission [11]. Other biological channels have been analyzed to some extent in Ref. [18], which crucially differs from the setup below in that we consider a population code so that information is contained in the frequency of alleles in the population; and in Ref. [19], they tackle how much entropy is added to the genome, rather than how much information is shared between entropic environments and a population of organisms with subpopulations that face different environments. It is also, therefore, of separate theoretical interest to calculate the channel capacity of a standard evolutionary channel based on the Wright-Fisher model.

Cultural evolution, too, usually implies mutation, or innovation [20]. By understanding how evolution is structured either for or not for information transfer by understanding mutation rate variability, we can understand how cultural evolution might be structured similarly or differently.

The manuscript is as follows. In Section 2, we describe channel capacity, the Blahut-Arimoto algorithm, and the Wright-Fisher model. In Section 3, we describe exact and lower bounds on the channel capacity of the Wright-Fisher evolutionary channel. More importantly, we ascertain if the observed distribution of mutation rates is aligned with maximal information transfer. In Section 4, we conclude by discussing experiments that can unravel if the proposed normative information-theoretic explanation for mutation rate variability is correct.

## 2. Background

In this study, we want to determine both the maximum information transferred between the environment (as coded by the mutation rate and selection coefficient) and the population’s allele frequencies and how Nature can achieve that maximum information transfer.

For organisms, information transfer occurs from generation to generation. The information is packed within DNA and transferred to the next generation, and this information transfer is affected by the environment. According to “survival of the fittest”, the genes transferred ensure the highest survivability rate in an environment. The information transfer we are concerned with is not the information transfer from generation to generation, but the overall information transfer from the environment (as coded by mutation rates and selection coefficients) to the population’s allele frequencies over time.

In order to achieve these goals, we will:Define our method of measuring the information transferred between the environment (as coded by mutation rates and selection coefficients) and allele frequencies.Describe the method for determining the maximum information transfer.Choose a simple and standard evolutionary model that describes a population of organisms and the environmental influences on the population.

### 2.1. Mutual Information

How can we measure the amount of information transferred from environment (as coded by the mutation rate and selection coefficient) to allele frequencies? We turn to Claude E. Shannon’s information theory. In his paper *A Mathematical Theory of Communication* [21], information theory is developed by conceptualizing communication systems such as electrical circuitry, telecommunication, or everyday human communication in the same framework. A communication system can be broken down into the following: the source of information, an encoder, a channel, a decoder, a receiver, and a destination. The *information source* is the sender of a message or messages. A message can be a letter, a word, or really any combination of symbols. The *encoder* is a device or any mechanism that converts the message into a signal suitable for the channel. The *channel* is the medium in which the message passes through in order to reach the receiver. The *decoder* is the opposite of an encoder. It is a device or mechanism that converts the signal back into the original message. The *destination* is the end-point. The message has finally reached its goal. Throughout the signal’s path it will encounter noise. *Noise* is a distortion that affects the message at any point in communication. Noise can occur at the transmitter if there is an error when the message is converted into a signal. In the channel, noise occurs when an external force (outside of the system) interrupts the signal. In the decoder, the signal does not convert back to the original message. For our system, the noise in the channel is fixed, and we attempt to encode the signal so that it can maximally transmit information through the channel.

Now that we have the foundation for finding the information transferred, how can we compute the amount? Let us consider a general communication system with a sender and a receiver. Suppose that there is a set of possible messages $\chi =[x_{0},...,x_{n}]$ that correspond to the random variable *X* with realizations *x* who have a probability distribution p(x). We know the set of possible messages, but we do not know at each iteration what message was sent. Therefore, we can define the uncertainty of the random variable as:(1)$$H\left[X\right]=-\sum_{x}^{n}p\left(x\right)\log p\left(x\right)$$ In information theory, Shannon defines *H* as a measure of information produced. In order to see this, let us consider a coin. A fair coin has the probability p=1/2 of landing on heads. We would not be able to reliably choose one outcome over the other. The entropy of the fair coin comes out to H=1 bit. However, let us now consider a weighted coin with probability p=1−q landing on heads where *q* is the probability of landing tails. We can now predict that the side weighted in favor of it will land more often. When we choose any value for q, we see that $H_{weighted}<H_{fair}$. The less certain about an event, the more information we need to describe the event. In contrast, if we know what outcome is most likely, then we need less information to describe the event.

Thus far, we only know the information in the original signal. In order to find the measure for information transferred, we need to consider noise in the channel. As mentioned, noise creates disturbances. In the channel, we cannot be certain of what these disturbances are. However, we know what message the decoder receives. If we know the message received, then we can compute the remaining uncertainty after we receive the message sent. Let us suppose that we know the message received *Y*, we can define the uncertainty of random variable *X* when we know random variable *Y* with condition probability distribution p(x|y) as the conditional entropy:(2)$$H\left[X|Y\right]=-\sum_{x,y}^{}p\left(y\right)p\left(x|y\right)\log p\left(x|y\right).$$ Let us get an intuitive sense of what the conditional entropy tells us. If conditional entropy is zero, then we are certain that the message received is the original message sent. This scenario entails that we can confirm both messages are identical. Hence, why there is no uncertainty. If conditional entropy is greater than zero, then there is some uncertainty. Perhaps the decoder made an error or the signal was disrupted as it traveled to the decoder. Either way we are no longer certain if the message received is identical to the original message. The smaller the conditional entropy, the less uncertain we are. The larger the conditional entropy, the more uncertain we are. With this intuition of conditional entropy, we can now find a measure for the information transferred.

In Shannon’s seminal thesis [21], he proposes that if 1000 bits are transferred and 10 bits are lost due to noise, then the information transfer would just be the bits transferred minus expected errors (990 bits). Because we lack the knowledge of the errors, we focus on what we know. The information transferred can be defined both as the number of bits transferred minus the number of bits lost to noise, and as as the reduction in the uncertainty about what was originally sent. Shannon defines this difference as *mutual information* and is denoted as:(3)I[X;Y]=H[X]−H[X|Y]=H[Y]−H[Y|X]. In order to understand mutual information, let us consider two cases: when mutual information is zero and when it is greater than zero. When mutual information is zero, then the message received and message sent are independent of one another. When mutual information is large, the channel is nearly noiseless, and nearly all information about the original signal is retained. The evolutionary channel will be somewhere in between these two extremes.

If *X* is a continuous random variable, then probability distributions are replaced by probability density functions.

### 2.2. Channel Capacity and Blahut-Arimoto Algorithm

In the previous section we found a measure for the information transferred through a channel. What if we were to shape the input so that as much information was transferred as possible? Shannon’s second theorem shows that the maximum rate of information transfer is the channel capacity:(4)$$C=\underset{p(x)}{\sup}I\left[X;Y\right]$$ According to Shannon’s second theorem, if the rate of information transfer is greater than the channel capacity, then there is a lot of error in decoding the information received. Minimal, vanishingly small error occurs when information is transferred below the channel capacity.

The Blahut-Arimoto algorithm will be our method of numerically computing the channel capacity. The Blahut-Arimoto algorithm is an optimization method for finding the channel capacity *C*. In Shannon’s definition of channel capacity, we are finding the supremum over p(x). This entails that p(y|x) is a fixed value. The Blahut-Arimoto uses an alternating maximization of I[X;Y] over p(x) and p(x|y), where p(y|x) is still fixed, so that p(x|y) is now a parameter. Convergence of the algorithm was proven by Csiszar and Tusnady [22].

With this in mind we can define the steps of the Blahut-Arimoto algorithm. The algorithm starts with a random p(x|y). Next, it computes p(x) using the the formula:(5)$$p\left(x\right)=\frac{\exp(\sum_{y}p\left(y|x\right)\log p\left(x|y\right))}{\sum_{x}\exp\left(\sum_{y}p\left(y|x\right)logp\left(x|y\right)\right)}$$ After this step, the algorithm alternates to compute for a new p(x|y) using the formula:(6)$$p\left(x|y\right)=\frac{p(x)\log p(y|x)}{\sum_{x}p\left(x\right)p\left(y|x\right)}$$ Next the algorithm will determine if channel capacity has been met by iterating Equations (Equation 5) and (Equation 6). If it did not reach the maximum channel capacity, it will continue to alternate Equations (Equation 5) and (Equation 6) until it finds the optimal input probability distribution $p^{*}\left(x\right)$.

After completing the computation, the Blahut-Arimoto algorithm is able to provide a channel capacity and an optimal input probability distribution $p^{*}\left(x\right)$ that maximizes mutual information.

If *X* is a continuous random variable, then probability distributions are replaced by probability density functions.

### 2.3. Wright-Fisher Model

In order to use these ideas to study the mutual information between environment (as coded by the mutation rates and selection coefficients) and a population’s allele frequencies, we need a model to describe frequency of the population *f* with allele *A* and allele *B* at generation *t*. The Wright-Fisher model aims to consider the forces of random genetic drift, mutation, and selection pressure that affect an allele’s frequency over generations. Each organism in generation *t* is replaced by the offspring in generation t+1. For this model, we are allowing allele *A* to be beneficial or deleterious relative to allele *B*. To avoid switching the label of the allele in the middle of the calculation, we allow instead *s* to be negative or positive.

Let’s suppose that for a fixed population size *N*, the fraction of the population with allele *A* is *f*. The population will under go through two processes in order to create the new generation: reproduction with selection, and mutation. Let’s consider the reproduction stage to be a half step before generation t+1 is created, $t+\frac{1}{2}$. In the reproduction stage, the probability of allele *A*’s reproduction is influenced by the previous generations frequency *f* of allele *A* and selection pressure. The selection pressure determines which allele is beneficial to survive and reproduce in the environment. This selection pressure can be quantified by the selection coefficient *s*. The reproduction of allele *A* is set by the probability $\frac{(1+s)f}{(1+s)f+1-f}$. When s>0, the allele *A* is said to be beneficial. Meanwhile, s<0 indicates a deleterious allele. Now let us consider the second stage to produce a new generation: mutation. The frequency of allele *A* for generation t+1, which undergoes mutation, is dependent on the frequency of allele *A* for generation $t+\frac{1}{2}$. The mutation rate can also influence frequency of allele *A* in generation t+1. The mutation rate μ is the probability that allele *A* changes to allele *B* or vice versa.

We want to consider the diffusion approximation which is valid when *N* is large meaning s→s/N and μ→μ/N. As such, *s* and μ can now have arbitrarily large magnitude. For instance, a μ of 80 actually corresponds to a mutation rate of 80/N, and in the limit that N→∞ which we take here, the mutation rate remains between 0 and 1. According, to the Central Limit Theorem, as *N* gets larger, the binomial random sampling is well approximated by a normal distribution. Therefore, if we combine both the selection and mutation stages, the Wright-Fisher model provides us with the following relation:(7)$$f_{t+1}=f_{t}+\frac{\mu }{N}\left(1-2f_{t}\right)+\frac{sf_{t}}{N}\left(1-f_{t}\right)+\frac{1}{\sqrt{N}}\sqrt{f_{t}\left(1-f_{t}\right)}Z_{t}$$
where $Z_{t}$ is zero-mean, unit-variance Gaussian noise and $f_{t}$ is the frequency of allele *A* at generation *t*. This equation tells us what the frequency of allele *A* will look like after going through one generation of reproduction with selection and mutation.

We used the above frequency equation $f_{t+1}$ in order to create the Langevin equation with time dt=1/N being a generation. Thus, we are working with the stochastic differential equation
(8)$$df=\left(\mu \left(1-2f_{t}\right)+sf_{t}\left(1-f_{t}\right)\right)dt+\sqrt{f_{t}\left(1-f_{t}\right)}d\eta ,$$
where dη is Gaussian noise with zero mean and variance dt. The Langevin equation maps to a Fokker-Planck equation. The Fokker-Planck equation is used to study the time evolution of a probability density function. In our case, it is the probability of producing a frequency dependent on the selection coefficient and mutation rates.
(9)$$\frac{\partial \rho (f|s,\mu )}{\partial t}=-\frac{\partial }{\partial f}\left(\mu \left(1-2f\right)+sf\left(1-f\right)\right)\rho \left(f|s,\mu \right))+\frac{\partial^{2}}{\partial f^{2}}\left(\frac{1}{2}f\left(1-f\right)\rho \left(f|s,\mu \right)\right)$$
where the drift term is μ(1−2f)+sf(1−f), so that mutation drives one towards a mixed population f=1/2 and selection drives one towards having only one allele, and the diffusion term is $\frac{1}{2}f\left(1-f\right)$. When μ=0, the probability favors the frequency of allele *A* to be around f=1 if s>0 and probability favors the frequency of allele *A* to be around f=0 if s<0. When s=0 and μ>1/2, the probability promotes the frequency of allele *A* to be near 1/2. When s=0 and μ=1/2, we find that the probability of seeing a particular frequency of allele *A* is uniform. When s=0 and μ=0, diffusion is the only force driving fixation with equal probability of either allele fixating. These limiting cases can be seen in Figure 1.

We solve for the steady state solution for the Fokker-Planck equation. The solution provides the probability of getting a frequency *f* for generation $t_{steady}$ given selection coefficient and mutation is known. This solution can be derived by noting that the steady-state solution satisfies
(10)$$0=\frac{d}{df}\left(-\left(\mu \left(1-2f\right)+sf\left(1-f\right)\right)\rho \left(f|s,\mu \right)+\frac{d}{df}\left(\frac{1}{2}f\left(1-f\right)\rho \left(f|s,\mu \right)\right)\right)$$
and so can be solved by setting the expression in parentheses, or the probability current, to 0:(11)$$0=-\left(\mu \left(1-2f\right)+sf\left(1-f\right)\right)\rho \left(f|s,\mu \right)+\frac{d}{df}\left(\frac{1}{2}f\left(1-f\right)\rho \left(f|s,\mu \right)\right)$$
which leads to
(12)$$\begin{array}{ccc} \left(\mu (1-2f)+sf(1-f))\rho (f|s,\mu \right) & = & \frac{1}{2}\left(1-2f\right)\rho \left(f|s,\mu \right)+\frac{1}{2}f\left(1-f\right)\frac{d\rho (f|s,\mu )}{df} \end{array}$$
(13)$$\begin{array}{ccc} \left(2s+\frac{\left(2\mu -1)(1-2f\right)}{f(1-f)}\right)df & = & \frac{d\rho (f|s,\mu )}{\rho (f|s,\mu )} \end{array}$$ Some straightforward integration gives a steady state solution as follows:(14)$$\rho \left(f|s,\mu \right)=\frac{{\left(f(1-f)\right)}^{2\mu -1}e^{2sf}}{\int_{0}^{1}{\left(f^{\prime }\left(1-f^{\prime }\right)\right)}^{2\mu -1}e^{2sf^{\prime }}df^{\prime }}$$
where the denominator normalizes the probability density function.

This probability density function allow us to compute the channel capacity and optimal probability distribution of the Blahut-Arimoto algorithm:(15)$$p^{*}\left(s,\mu \right)=\arg\max_{p(s,\mu )}I\left[f;s,\mu \right]$$ The relationship states that Nature is maximizing information if experimental probability distribution over mutation rates and selection coefficients matches the optimal probability distribution over mutation rates and selection coefficients. We consider the case that a large population of organisms is divided into subpopulations of size *N* that all feel different s,μ combinations, so that there can be repeated channel uses with different inputs without having to worry about transients associated with switching from one combination of s,μ to the next.

## 3. Results

In this section, we analyze the channel capacity of the environment-population’s allele frequency channel modeled by the Wright-Fisher model Figure 2. The Blahut-Arimoto algorithm will provide us the channel capacity and optimal probability distributions.

Analysis will be repeated for two subsections. One will focus on a special case where no mutation is present in the environment, to build intuition. The second case will focus on a typical environment with selection and mutation present.

### 3.1. No Mutation

We will present results from a special case in which the population experiences no mutation. When there is no mutation, one allele fixates [23]. The probability that allele *A* fixates is given by $p\left(f=1|s\right)=\frac{e^{2s}}{1+e^{2s}}$, and the probability that allele *B* fixates is given by $p\left(f=0|s\right)=\frac{1}{1+e^{2s}}$. Our goal is to determine the optimal probability distribution over selection coefficients and the channel capacity.

First, we focused on determining the optimal probability distribution over selection. We utilized the Blahut-Arimoto algorithm with our conditional probability distribution from above as the input. In our simulation, the possible selection coefficients ranged from −8 to 8 with 1000 values within this range, and was ran for a substantial amount of iterations to allow for convergence. The Blahut-Arimoto algorithm produced the optimal probability distribution function over selection in Figure 3; see Appendix B. The figure suggests that the environment-population channel maximizes information when heavy weight is placed on large-magnitude selection coefficients.

Based on the results from the Blahut-Arimoto algorithm, we can analytically determine the channel capacity. We found a channel capacity of 1 bit when there is no mutation for the environment-population’s allele frequency channel in Lemma 1.

**Lemma** **1.**
*Let I[f;s,μ] be the mutual information transferred between the environmental influences and the population’s allele frequency, and let ρ(s) denote the input distribution. If the mutation rate is equal to zero, then we find that the optimal input distribution of selection coefficients places equal weight on infinitely negative and positive selection coefficients and the maximum mutual information (or channel capacity) is 1 bit.*


**Proof.** Without mutation, there is fixation, meaning that there is only support for ρ(f) on f=0 and f=1. That means that ρ(f) becomes p(f), a probability distribution, with support on two points– f=0 and f=1. Hence,
(16)I[f;s]≤H[f]≤1bit. We can achieve this limit of 1 bit by placing equal weight on s=−∞ and s=∞. Then,
(17)H[f|s]=0bits
and
(18)$$\begin{array}{cc} p(f=0) & =\sum_{s}p\left(s\right)p\left(f=0|s\right)=\frac{1}{2} \end{array}$$
(19)$$\begin{array}{cc} H[f] & =1\;\mathrm{bit}. \end{array}$$ Hence, with this input distribution,
(20)I[f;s]=H[f]−H[f|s]=1bit−0bits=1bit,
and channel capacity must be 1 bit achieved by an input distribution that places equal weight on s=−∞ and s=∞. □

We can extend Lemma 1 for bounded selection coefficients. This is more biologically realistic in that one cannot have infinite selection coefficients, but the channel capacity diminishes slightly.

**Lemma** **2.**
*Let I[f;s,μ] be the mutual information transferred between the environmental influences and a population’s allele frequencies, and let ρ(s) denote the input distribution over selection coefficients. In addition the range of selection coefficients range from $-s_{max}$ to $s_{max}$. If the mutation rate is equal to zero, then we find that the optimal input distribution of selection coefficients places equal weight on the positive bound $s_{max}$ and negative bound $-s_{max}$ and the maximum mutual information is*

(21)
$$I\left[f;s\right]=1-H_{b}\left(\frac{1}{1+e^{2s_{max}}}\right)$$



**Proof.** Let *s* be any selection coefficient within the range $\left[-s_{max},s_{max}\right]$, a realization of the random variable *S*. Without mutation, there is fixation. In order to find the maximum mutual information, or channel capacity, it is enough to find when H[f] is maximized and H[f|s] is minimized. The former occurs when $p\left(f=0\right)=\frac{1}{2}$, which can be achieved by a symmetric p(s). (Recall that *f* is now a discrete random variable when there is no mutation with support on f=0 and f=1. To minimize H[f|s], we find $s=\pm s_{max}$, as the probability of fixating the less likely allele in the population decreases monotonically away from s=0. Together, these two criteria imply that $p\left(s=\pm s_{max}\right)=\frac{1}{2}$ and that the channel capacity is as stated in the lemma statement, where $H_{b}\left(x\right)=-x\log_{2}x-\left(1-x\right)\log_{2}\left(1-x\right)$. □

### 3.2. With Mutation

When there was both selection and mutation, the Blahut-Arimoto algorithm did not converge to an optimal probability distribution in the compute time allotted—see Appendix B—and so we analytically found a lower bound on the channel capacity.

Imagine that the support of the probability distribution over selection coefficients and mutation rates is only on: μ=0.5, s=0 so that a highly entropic allele frequency distribution is experienced, i.e., there is a uniform distribution of allele frequencies as seen in Figure 1 middle panel; s→±∞, μ=0 as per the previous subsection; and μ→∞, s=0 so that a completely mixed allele frequency distribution is seen, i.e., there is a peaked distribution centered at f=0.5 as seen in Figure 1 top-middle panel. It turns out that this probability distribution leads to a mutual information between environmental influences and population’s allele frequencies of 2 bits or 1.4 nats.

**Lemma** **3.**
*The channel capacity of the Wright-Fisher evolutionary channel is lower-bounded by 2 bits.*


**Proof.** Since the channel capacity is lower-bounded by the mutual information for any distribution of inputs, we can choose a distribution, calculate the mutual information, and lower bound the channel capacity. We choose a discretely-supported distribution:
(22)$$\begin{array}{cc} p(s\to +\infty ,\mu =0) & =\frac{1}{4} \end{array}$$
(23)$$\begin{array}{cc} p(s\to -\infty ,\mu =0) & =\frac{1}{4} \end{array}$$
(24)$$\begin{array}{cc} p(s=0,\mu =0.5) & =\frac{1}{4} \end{array}$$
(25)$$\begin{array}{cc} p(s=0,\mu \to \infty ) & =\frac{1}{4}. \end{array}$$ One can also start with arbitrary constants for the probabilities, but this combination of constants happens to optimize the mutual information. Using the ϵ-entropy definition for mixed random variables [4,24], as *f* now is, we have
(26)$$\begin{array}{cc} I[f;s,\mu ] & =H[f]-H[f|s,\mu ] \end{array}$$
(27)$$\begin{array}{cc} \; & =\lim_{\epsilon \to 0}\left(3\left(\frac{1}{4}+\frac{1}{4}\epsilon \right)\log\frac{1}{\frac{1}{4}+\frac{\epsilon }{4}}+\left(\frac{1}{4}-\frac{3}{4}\epsilon \right)\log\frac{4}{\epsilon }+\left(\frac{1}{4}-\frac{3}{4}\epsilon \right)\log_{2}\epsilon \right) \end{array}$$
(28)$$\begin{array}{cc} & =\log4. \end{array}$$ In bits, this is 2 bits; in nats, this is 1.4 nats. See Appendix A for more detail. □

The interpretation of this probability distribution, which may be non-optimal, is that we have placed weight on four (s,μ) combinations that can be completely read out by looking at the allele frequencies *f*. If the allele frequencies are only 1 or 0, then s=±∞, μ=0 is in the environment; if the allele frequency is uniformly distributed, then we have s=0,μ=0.5; and if the allele frequency is placed squarely at $f=\frac{1}{2}$, then s=0 and μ→∞.

Note that our lower bound on the channel capacity is bounded above by the logarithm of the number of different possible s,μ combinations. This is no accident, because $I\left[f;s,\mu \right]\leq H\left[s,\mu \right]\leq \log_{2}N$. If these bounds were loose, we could not be sure that logN was a lower bound on the true channel capacity, as the maximization of both sides over p(s,μ) (which would lead to the channel capacity on the left-hand side) might break the chain of inequalities. To achieve equality and therefore assure ourselves that logN was truly a lower bound on channel capacity, we need that H[s,μ|f]=0– in other words, if one sees the allele frequency distribution, that there is no doubt in one’s mind as to what selection benefits and mutation rates are there. One might reasonably ask, can we add another s,μ point to the input and still have H[s,μ|f]=0? The answer, it seems, is surprisingly, no. The first hint that the answer is no comes from numerical simulations with the Blahut-Arimoto algorithm, which show that our lower bound beats what is found by the algorithm, not just for the case of selection and mutation, but for the case of mutation but no selection. The second hint comes from the thought experiment of trying to add another s,μ point that doesn’t result in a highly peaked distribution for ρ(f|s,μ). If one sees that f=1/4, say, is present, the most one can say is that some s,μ combination besides s→±∞,μ=0 and s=0,μ→∞ is present. For H[s,μ|f] to remain 0, one needs there to be only one s,μ combination that might have produced f=1/4.

## 4. Discussion

We calculated a lower bound on the channel capacity of an evolutionary channel from environmental factors to an experimentalist observing populations of organisms changing allele frequencies from generation to generation to be 2 bits. Simultaneously, we utilized the Blahut-Arimoto algorithm and analytical intuitions to find approximations to the probability distributions over selection coefficients and mutation rates that maximally ease the process of selection benefit and mutation rate inference for the watching experimentalist. The corresponding distribution over allele frequencies is highly peaked about a perfectly mixed population and two populations representing the two extremes, in which allele *A* and allele *B* take over the population, but also allows for a uniform smearing of allele frequencies that represent a mutation rate of 1/2N and no selection benefit. This, then, represents the environmental influences that would be easiest to infer from allele frequencies.

The channel capacity of two bits per allele is relatively high for signal transduction pathways that have been previously studied. To our knowledge, this is the first calculation of lower bounds on the channel capacity of an evolutionary channel. Although our guessed probability distributions over selection coefficients and mutation rates are likely not optimal, the Blahut-Arimoto algorithm provided some support for our channel capacity lower bound being nearly correct.

We hope future work can address fluctuating environments [6], for example, and move from discussing allele frequencies to discussing phenotypes so that perhaps a principle of achieving channel capacity can be utilized to understand evolving populations. Another important extension to this work will be to restrict the setup to having mutation rate evolve in a way that increases the fitness of the population. We expect that most adaptation rules will drive mutation rates away from the special mutation rate that allows for a uniform smearing of allele frequencies (μ=1/2N) and will drive them towards mutation rates that result in peaked distributions for allele frequencies. In such cases, constraining the possible rules for changes to mutation rates over time will lower the channel capacity to $\log_{2}\left(3\right)=1.6$ bits.

## Figures and Tables

**Figure 1 entropy-25-00615-f001:**
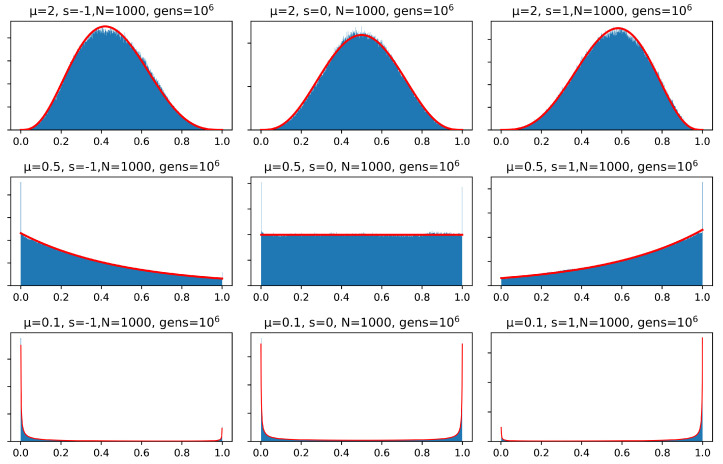
Steady state distribution Equation (Equation 14) (red line) and normalized frequency histogram (blue area). The frequencies were simulated through binomial sampling of a population size $N=10^{3}$ and $10^{6}$ generations in the diffusion approximation such that μ→μ/N and s→s/N for various limiting cases. From top to bottom we have three cases of μ=2,1/2, and 0.1. From left to right we have three cases of s=−1,0, and 1. In the bottom row, when μ is very close to zero, there is heavy weight on f=0 and/or f=1 which leads to infinitely large peaks (peaks are finite in figure for visualization purposes).

**Figure 2 entropy-25-00615-f002:**
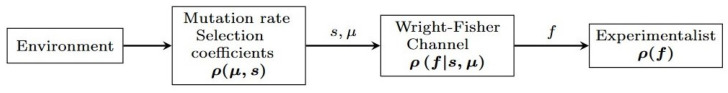
Mutator alleles and transcription factors alter the mutation rate and selection coefficients of an asexual organism based on information about the environmental influences. Through evolutionary processes modeled by the Wright-Fisher model, the frequency of organisms with allele *A* information about the environment can be interpreted by an experimentalist.

**Figure 3 entropy-25-00615-f003:**
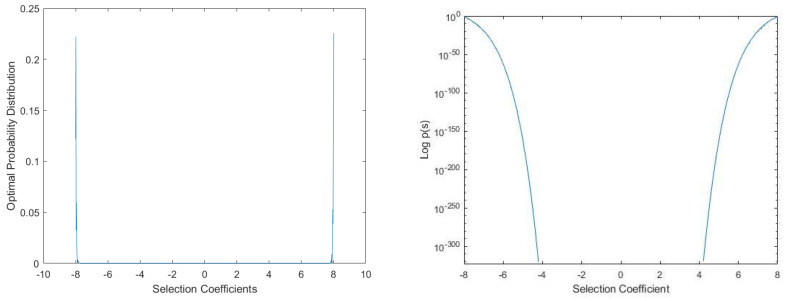
(**Left**): Optimal probability density function of source symbols, $\rho^{*}\left(s\right)$ vs. selection coefficients *s* for mutation rate equal to zero. High weight is placed on the minimal and maximal selection coefficients to maximize information transfer from environmental influences to population’s allele frequencies. (**Right**): we have the optimal distribution over *s* in log-scale on the y-axis.

## Data Availability

Not applicable.

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
