# Peer review of "How Well Can We Infer Selection Benefits and Mutation Rates from Allele Frequencies?"

_entropy, 2023, doi:10.3390/e25040615_

Round 1

Author Response

We hope that the changes made have answered the concerns. This was a particularly wonderful review in that it really helped us think through several issues, and we'd like to thank the referee.

Reviewer 2 Report

In this contribution, the authors attempt to calculate the amount of information that a population has acquired from the environment and compare it to the maximal value (the channel capacity), in order to test whether evolution has driven the information acquisition rate toward the maximal value.

I need to point out a few facts. This is a binary one allele-model (alleles A or B). One allele has fitness 1, the other 1+s. Therefore, s can be large (strong selection), but it cannot be negative. But Fig. 1 shows results for s<0, up to s=-80. In other places, the authors refer to a limit s-> minus infinity. This does not make any sense. 

Disregarding this for a moment, it is clear that the maximal amount of information that can be acquired here is 1 bit: the single bit that is the single binary locus. Where do these numbers with information larger than 1 bit come from? It appears that the authors now construct a different model: one where there is not a single peak with advantage s, but rather a landscape with peaks of all sizes (I'm ignoring the negative s here). Then they ask what the probability distribution is given this landscape with different-sized peaks, and find that (for vanishing mutation rates) the population will fix at the largest s. That is trivial. Still, I don't know where the fixation probability in line 173 comes from. It is not Kimura's (which is 1-e^(-2s)) and it is not Haldane's, which is (2s+s^2)/(1+s)^2.

The authors then go on to discuss finite mutation rates. I first need to point out that mutation rates are given in terms of the number of mutations per locus per generation. Since there is only a single locus, the maximal mutation rate is 1: this means that the locus is flipped every generation. A mutation rate of 20 is nonsensical. It does not matter if I flip the locus one time, ten times, or 100 times between Wright-Fisher generations. The authors then go on to lower-bound an information, based on this probability distribution over s (which is not the probability distribution of frequencies p(f) discussed in the earlier section). The latter would be appropriate to discuss information acquired (and this is of course bounded by 1 bit). This other distribution is on a continuous variable bounded by zero and infinity. Its entropy is infinite. But the authors want to lower bound it, so they introduce a discretized support. This is arbitrary, of course. When introducing four support values, they get a lower bound of log 4. When introducing 3 support values, they get log(3). None of this has anything to do with the information transferred. However, they do make this comparison and find that they don't agree. They then compare to the empirical distribution of mutation rates (not selective advantages, since we know what this would look like) in yeast, and find this distribution to be very different. This should not be surprising, since the model the authors are presenting (where infinite mutation rates can exist) has nothing to do with reality. 

I thought the paper was going in an entirely different direction when the authors were calculating the equilibrium distribution of the Wright Fisher process using the Fokker-Planck equation, and comparing that to the maximum amount of information that the channel can have (1 bit). They would have found that the information depends on the mutation rate, and approaches one bit as mu--> 0. However, this is not what the authors did. As mu-->1, the information becomes zero because the noise is maximal and no information can be transferred in such a channel. This is the limit of the "useless channel". 

Instead, they focus on a probability distribution where s is a variable (as opposed to the fixed value in (9), where f is the variable). These are completely different scenarios that cannot be compared. We are not dealing with a Fokker-Planck equation with derivatives d/ds. 

That the authors find a "lack of a match" therefore is not surprising, as they are comparing two entirely different things.

Author Response

We hope that the changes made clarify the issues and make the paper suitable for publication.

Reviewer 3 Report

The paper considers two deep and interesting questions: what determines mutation rate? and how to quantify the amount of information that organisms acquire about their environment.

However, my opinion is that the paper is at the beginning of attempting to answer these questions.  The first bound given is not surprising. The second bound is hard to interpret, since the set of "messages" include variations in mutation rate: discriminating between "messages" sent with different mutation rates - if variation in mutation rate is in some sense in the "message" - is surely not determining information about the environment?

I would direct the authors' attention to the (very) recent paper 

Accumulation and maintenance of information in evolution, Hledik, Barton, and Tkacik
Proceedings of the National Academy of Sciences, 119,36, e2123152119, 2022

and some previous work cited in that paper.

Selective breeding analysed as a communication channel: channel capacity as a fundamental limit on adaptive complexity Watkins, Chris 2008 10th International Symposium on Symbolic and Numeric Algorithms for Scientific Computing pp514--518, 2008, IEEE
These papers describe (slightly different) communication channels between environment and genome(s) which seem at least related to the communication channels that the authors propose.

Author Response

Thank you so much for the references and questions. We hope the paper is now more obviously a substantial step forward.

Round 2

Reviewer 1 Report

The authors have addressed my concerns and have made significant improvements to the manuscript. I accept the current version.

Author Response

Thank you for your comments

Reviewer 2 Report

The authors have provided a revised version of their manuscript that clarifies some misunderstandings. But some others remain.

They write that in the Wright-Fisher model with one allele, the fitness advantage can be positive or negative. This is true: if it is positive for one of the alleles, it will be negative for the other. What this implies is that the situation is completely symmetric: we only have to treat one of the s (positive or negative). Fig. 3, then, makes no sense: it shows the probability distribution of "optimal" selection coefficients. But this should be the distribution of fixation probabilities against the *same* background. If you have a benefit s over a wild-type with fitness 1, then your fitness is 1+s. The probability of fixation of an allele with fitness 1-s is zero. You can see the immediately from Eq (10). 

Of course you have maximal information about a population where the frequency of the introduced type is zero. You can get that if the information measure you define is based on the frequencies of the type, \rho(f).But that is not a measure of information that the population has acquired. It is a measure of how well you can predict whether one or the other type dominates the population. You, as the one who observes the population. If the population is uniform (middle panel of Fig. 1) then your uncertainty should be largest. And in fact it should be very large, as this should be given by a differential entropy.  

The authors go on to say that if an allele fixes then the information should be largest (one bit): yes: now we can easily predict who went to fixation! But this has nothing to do with the information acquired by the environment!  Even more, I don't understand how the authors can argue that I[f,s,\mu]=1bit

They do so by writing that I(f,s)=H(f)-H(f|s). This is a very strange formulation, when it comes to information theory. We can argue that f is a random variable. If it is not conditioned, then the entropy of that variable should be maximal, given some constraints in the system. There are no constraints except normalization, so H)f) is maximal when f is uniform. But this is not one bit! In fact, the differential entropy of a uniform distribution over the interval (0,1) [which is what the middle panel of Fig. 1 is), is zero!

If we choose to discretize the differential entropy with a discretization \Delta f, then the entropy is log(\Delta f). This is one bit only if \Delta f=0.5. That's not a good discretization, but in a sense a lot of the author's arguments depend on introducing such a discretization, which implies that there really are only three different frequency distributions possible: f=0, f=1, and f=1/2. The first two are obvious cases of fixation. The last one can only happen if mutations essentially flip the two alleles back and forth every generation. This happens at \mu=0.5. 

But I haven't even talked about H(f|s). What is that supposed to mean? First, in information theory we discuss the correlation between two random variables using information. While f is such a variable, s is not. In this case, it makes no sense to say "given s", because s is always given. The authors later describe a situation where s and mu are thought to be random variables in an ensemble of populations. That can make sense, but now these random variables are not on the same level: giving s (picking one particular population from the ensemble with a given s) does not affect f. Or put another way: the frequency distribution rho(f) only makes sense is f s is given. The distribution without giving s would be an overlap of all possible distributions (such as those in Fig. 1). I don't know what this would look like, and I don't think the authors know either, in particular if the mutation rate is included as a variable. 

And I want to repeat my comment that \mu=1 is the highest possible mutation rate. In that case, I replace every allele with the opposing allele every generation. There is at no time a mixtures of alleles. It is a perfect channel in theses that information is not lost: you can always predict who is dominating the population. 

And this brings me to the most important criticism. While everything up to Eq. (10) is a perfectly reasonable (and from what I can tell, correct) derivation of the probability density of alleles frequencies in the steady-state limit, casting this in terms of an information-theoretic channel that quantifies the information that is acquired by a population from the environment is wrong. What this is (if done correctly, i.e., treating the unconditional entropy correctly by using a discretization of frequencies that is not 0.5 but much smaller) the information that an observer has about the population composition. If the observer knows that there is no mutation rate,*and* they know nothing about the allele fitness, they would have to have a prior saying that either of the two alleles could be fixed (H=1 bit). If they know that the background is 1 and the allele introduced has fitness 1+s, then they can say with certainty that this allele will fix (if N is large enough). As an aside, it is curious that this N dependence of the fixation probability does not appear in the equations, even though the Kimura formula for fixation has such an N dependence.

I will stop here (there is more to say, such as arbitrarily choosing a support of four points in the (s,mu) plane to bound the entropy. And using mu>1.)

None of what the authors do (running an algorithm to find the probability distribution that maximizes the information) makes any sense until the quantities they use are well-defined, and understood. 

I'm not saying this is all hopeless. It just needs to be cast right. As I said, everything up to Eq. (10) is correct (I even checked that this distribution does indeed solve the stationary FP equation, which at first I didn't believe).

This work is about the predictability of evolutionary dynamics, not about information acquired by a population. 

Author Response

See comments in the attachment. We are worried that the entire setup does not make sense to you as an information theoretic setup, and if this communication manages to clear up the difficulties we have in understanding each other, it would be great to hear how we could alter the manuscript so as to better get these points across.

Round 3

Reviewer 2 Report

see attached file

Author Response

Thank you for your careful consideration. We greatly appreciate your help in revising the manuscript.

We have substantially revised our manuscript-- in particular, the purpose of why we are doing what we are doing-- due to your comments. I hope that this latest version is satisfactory.

In particular, we agree with all points made by the reviewer and only point out that:

-- the dt need not be there since d\eta has variance dt

-- if s of the allele is nonzero and mutation rate nonzero, only due to our renormalization, there is a nonzero probability of finding allele frequencies that are between 0 and 1

-- we have now added motivation for why we would want to maximize the mutual information for a different reason

-- we don't think we use the probability distribution over s for the probabilities of fixation

-- the argument for H(f|\mu=0) is not quite right, please see manuscript, in particular you have to choose p(s) so that f is equally likely to be 0 or 1 in the ensemble (imagine that p(s) is not symmetric for an instance of the opposite)

-- we tried to make an argument for why it must be \log 4, please see manuscript!

-- I would not say that \mu=0 leads to an easy to predict distribution, because despite having no differential entropy, it is a maximum entropy distribution (differential entropy being sufficiently different than normal entropy that the meaning of the entropy does not carry over)

-- we have taken the yeast system out, please see manuscript

-- we have now talked about the physicality of \mu=0.5.

Round 4

Reviewer 2 Report

This is a substantially changed version of the manuscript that addresses my most pressing concerns. Note that in section 2, the ms still refers to the task of "measuring information transfer from the environment to the population", which is now going to confuse the reader. I trust the authors will be able to spot ant other leftover confusing remarks. I do not need to see the manuscript again.  

Author Response

We have attempted to make all changes suggested.  We no longer refer to the information transfer from environment to organism, or even from environment to population.  We are careful to say environmental influences to population's allele frequencies.

Thank you!